# mTOR signaling regulates the morphology and migration of outer radial glia in developing human cortex

Madeline G Andrews[1,2†], Lakshmi Subramanian[1,2†]*, Arnold R Kriegstein[1,2]*

[1]Department of Neurology, University of California, San Francisco (UCSF), San Francisco, United States; [2]The Eli and Edythe Broad Center of Regeneration Medicine and Stem Cell Research, UCSF, San Francisco, United States

**Abstract** Outer radial glial (oRG) cells are a population of neural stem cells prevalent in the developing human cortex that contribute to its cellular diversity and evolutionary expansion. The mammalian Target of Rapamycin (mTOR) signaling pathway is active in human oRG cells. Mutations in mTOR pathway genes are linked to a variety of neurodevelopmental disorders and malformations of cortical development. We find that dysregulation of mTOR signaling specifically affects oRG cells, but not other progenitor types, by changing the actin cytoskeleton through the activity of the Rho-GTPase, CDC42. These effects change oRG cellular morphology, migration, and mitotic behavior, but do not affect proliferation or cell fate. Thus, mTOR signaling can regulate the architecture of the developing human cortex by maintaining the cytoskeletal organization of oRG cells and the radial glia scaffold. Our study provides insight into how mTOR dysregulation may contribute to neurodevelopmental disease.

*For correspondence:
lakshmi.subramanian.me@gmail.com (LS);
Arnold.Kriegstein@ucsf.edu (ARK)

†These authors contributed equally to this work

Competing interests: The authors declare that no competing interests exist.

## Introduction

The evolutionary expansion of the human cerebral cortex has been attributed to an increase in progenitor cell number and diversity (*Borrell and Götz, 2014*; *Lui et al., 2011*; *Rakic, 2009*). In particular, oRG cells, a predominant progenitor population in the developing human cortex, contribute extensively to cortical expansion and folding due, in part, to their morphology and unique mitotic behavior (*Hansen et al., 2010*; *Matsumoto et al., 2020*). oRG cells undergo mitotic somal translocation (MST) where the nucleus moves rapidly along the basal process towards the pial surface prior to mitosis (*Hansen et al., 2010*). The migration and MST behaviors of oRG cells depend greatly on the integrity of the basal process (*Betizeau et al., 2013*; *Kalebic et al., 2019*; *Ostrem et al., 2014*). oRG basal processes also function as a glial scaffold to support neuronal migration (*Gertz and Kriegstein, 2015*; *Nowakowski et al., 2016*). The specific morphology and dynamic behavior of oRG cells contribute to the proliferative expansion of the cortex and provide a scaffold upon which neuronal migration depends.

Recent studies have identified high levels of mTOR signaling in human oRG cells (*Nowakowski et al., 2017*; *Pollen et al., 2019*). Multiple neurodevelopmental disorders and malformations of cortical development are thought to arise due to the effects of mTOR signaling mutations on progenitor cells during fetal development (*Subramanian et al., 2019*). Because oRG cells are a major progenitor population during neurogenesis and show significant mTOR activity at this stage, they may be particularly vulnerable to the effects of dysregulated mTOR signaling during a critical developmental period. Recent studies ectopically hyperactivating the mTOR Complex 1 (C1) pathway in mouse models have identified phenotypic changes to neuronal size, morphology, and laminar position that resemble disease features of cortical malformations (*Hu et al., 2018*; *Lim et al., 2017*; *Nguyen et al., 2019*; *Park et al., 2018*). However, questions remain regarding the effects of mTOR

dysregulation on progenitor cells as well as relevant differences between human and mouse cortical development. oRG cells are a vital cortical stem cell population in humans, but they are much less abundant in the mouse cortex. While mTOR signaling can regulate aspects of cortical development in rodents, the role of this signaling pathway in human oRG cells is not yet understood.

We therefore investigated the functional role of mTOR signaling in oRG cells using both tissue samples and in vitro models of the developing human cortex. In addition to the culture of primary cortical tissue and dissociated cells, we utilized cortical organoids because of their tractability as a model system (*Andrews and Nowakowski, 2019*). Using these human model systems we observed that an appropriate level of mTOR activity is required to maintain oRG basal process morphology and migration behavior. As the basal processes of oRG cells contribute to the primary radial scaffold, particularly during the migration of upper layer cortical neurons (*Nowakowski et al., 2016*), balanced mTOR signaling is also crucial for the development of proper human cortical structure and organization.

## Results

### Changes in mTOR signaling disrupt the glial scaffold in primary cortical cultures and organoids

Mediators of mTOR signaling are expressed in discrete domains in the developing human cortex, in both primary cortical tissue and cortical organoid models (*Figure 1*). Activation of the mTORC1 pathway leads to the phosphorylation of S6 kinase, which can be used as a downstream readout of mTORC1 signaling activity (*Figure 1A*; *Saxton and Sabatini, 2017*). In the developing human cortex, oRG cells can be identified by the expression of HOPX (*Pollen et al., 2015*). Phosphorylated S6 (pS6) is detected in HOPX-expressing oRG cells, but not in other progenitor subtypes such as ventricular radial glia (vRG), truncated radial glia (tRG), and intermediate progenitors (IPCs), or in neurons (*Figure 1D–I*). To explore the role of mTOR signaling in oRG cells, we utilized organotypic slice cultures of primary cortical tissue from gestational weeks (GW) 15–19, the developmental period when oRG cells are rapidly expanding. This corresponds to the peak neurogenic stages in the developing human cortex. Cortical slice cultures were treated with rapamycin to inhibit, and BDNF (*Saxton and Sabatini, 2017*) or 3BDO (Sigma) (*Huang et al., 2017*) to hyperactivate, mTOR signaling (*Figure 2A*). In control slices, pS6 expression was most abundant in the outer subventricular zone (oSVZ), where most oRG cells reside. In rapamycin-treated slices, most of the pS6 expression was lost, while slices treated with BDNF/3BDO generally showed increased pS6, indicating activation of mTOR signaling (*Figure 2—figure supplement 1*).

Organotypic slice cultures were treated with a CMV::GFP adenovirus, that labels the progenitor cells and allows visualization of cortical radial architecture. GFP-labeled progenitor cell processes are visible in vehicle-treated sections as an organized scaffold of radially arranged basal fibers (*Figure 2C*). In contrast, both mTOR inhibited and hyper-activated slice cultures show a highly disrupted glial scaffold where the average length of radial fibers was significantly reduced (*Figure 2C*; n > 5 slices/group across >5 independent experiments, *Figure 2—figure supplement 1D*). Both pS6 expression and radial scaffold organization returned to near control levels in BDNF-treated slices after the addition of rapamycin, demonstrating that the observed effects are a result of specific changes to mTOR signaling and not due to off-target effects of BDNF (*Figure 2—figure supplement 1C*). The length of the basal process was significantly reduced in HOPX-expressing oRG cells following mTOR inhibition or hyper-activation (*Figure 2C*; n > 3 slices/group from >3 experiments). No effects on process length were observed in any other neural progenitor subtype following manipulation of mTOR signaling (*Figure 2—figure supplement 2*; n = 3 experiments). The orientation of the primary process towards the pial surface was also affected after mTOR manipulation. In control slices, the basal/primary processes of oRG cells are directed towards the pial surface. As a result, their angles with the ventricular surface are nearly perpendicular (61–90˚). In mTOR-manipulated slices, the average angle of the primary processes with the ventricular surface was reduced to mostly oblique (31–60˚) or almost parallel (0–30˚) (*Figure 2—figure supplement 2E*; n > 17 cells across three experiments).

Dissociated primary oRG cells retain their morphological characteristics in culture, with a single long primary fiber that defines their migration trajectory and MST behavior. Truncation of the

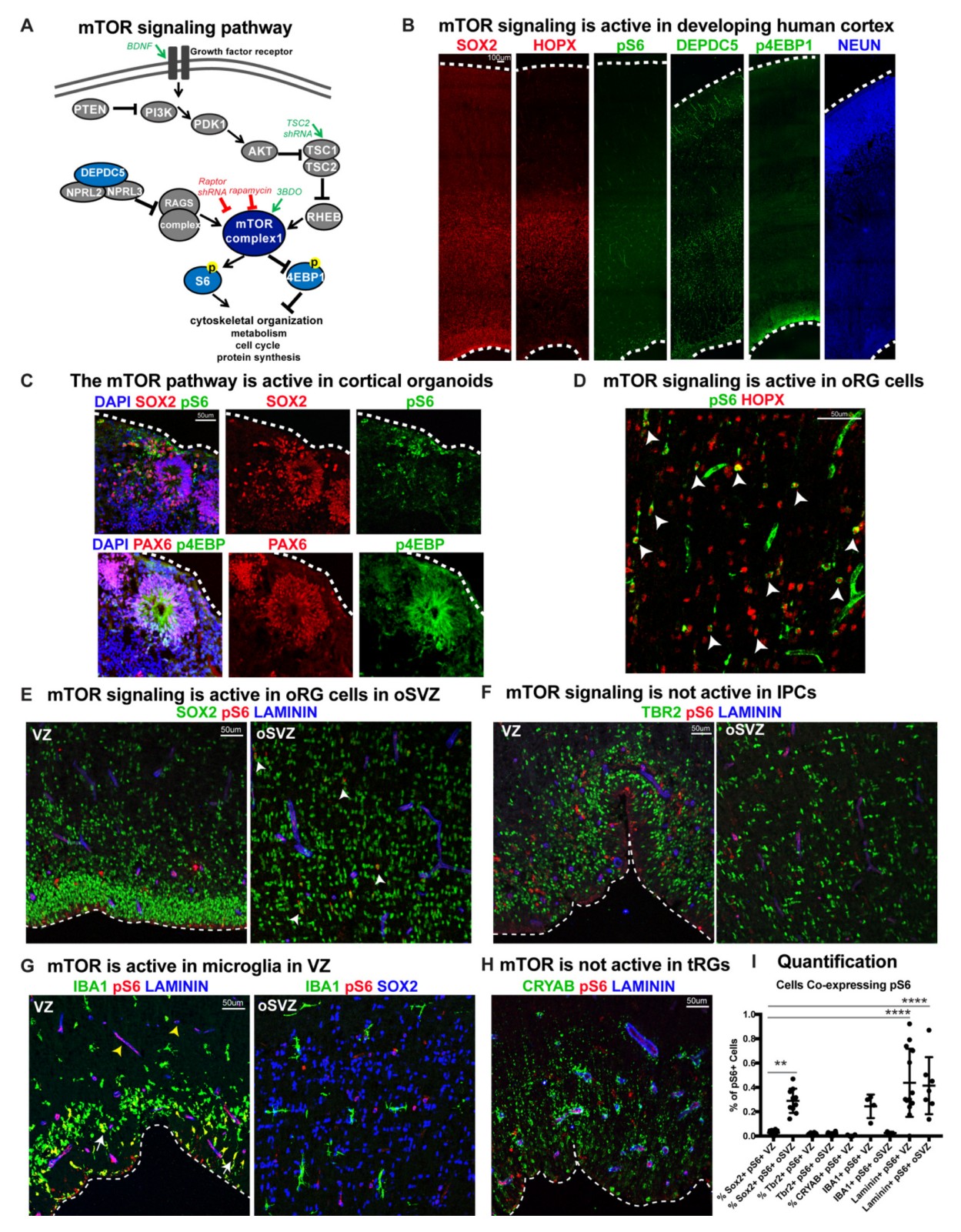

**Figure 1.** mTOR signaling is active in outer radial glial cells, but not other neural progenitor types, during human cortical development. (A) Simplified schematic of the mTORC1 signaling pathway. mTOR signaling has many mediators and regulates diverse processes throughout development. Experimental approaches to mTOR manipulation are included in the diagram. Green text indicates activation and red indicates inhibition. (B) Radial glia expressing SOX2, oRG cells expressing HOPX and neurons expressing NEUN are present during human corticogenesis at GW17-19. mTOR signaling is

*Figure 1 continued on next page*

*Figure 1 continued*

active in the cortex at this time, indicated by the presence of phosphorylated S6 (pS6) protein. Other mTOR pathway proteins like DEPDC5 and p4EBP1 are also expressed during this period. (C) mTOR pathway proteins are also expressed and active in the radial glia of eight week old cortical organoids. (D) pS6 is present in many, but not all, oRG cells in the oSVZ of the developing human cortex, indicating active mTOR signaling. White arrowheads indicate oRG cells co-expressing pS6 and HOPX. (E) pS6 is absent in SOX2+ vRGs in the VZ but present in laminin-expressing blood vessels. Of the neural progenitor populations, only SOX2+ oRG cells in the oSVZ co-express pS6 (white arrowheads). (F) pS6 is absent in TBR2+ IPCs in both the VZ and the oSVZ. (G) mTOR signaling is active in IBA1+ microglia (white arrows) and laminin+ blood vessels at the VZ (yellow arrowheads). However, in the oSVZ there is little mTOR activity in the microglia, but extensive co-labeling of SOX2+ oRG cells. (H) CRYAB+ tRGs do not express pS6. (I) Quantification of pS6+ cells co-labeled with each marker, SOX2, TBR2, CRYAB, LAMININ and IBA1, at the VZ or oSVZ expressed as a percentage of the total number of pS6+ cells. Significant mTOR activity is present only in the SOX2+ oRG cells in the oSVZ and in the laminin+ blood vessels in the VZ and oSVZ. (n = 28 sections from 3 individuals across eight independent experiments; D'Agostino-Pearson normality test: normally distributed; one way ANOVA with multiple comparisons: SOX2+ VZ vs SOX2+ oSVZ: **p<0.0031; SOX2 VZ vs Laminin VZ, Laminin oSVZ: ****p<0.0001; Scatter plot with individual data points, error bars indicate SD and the middle line is the mean).

The online version of this article includes the following source data for figure 1:

**Source data 1.**

---

primary fiber following mTOR inhibition or hyperactivation was also observed in dissociated progenitor cultures (*Figure 2E*; n > 12 HOPX+ cells/group across n = 5 experiments). Additionally, mTOR modulation resulted in other morphological changes to oRG cell processes including growth of multiple fibers extending from the cell body (*Figure 2E*, *Figure 2—figure supplement 2E*). These additional apical projections were most easily visualized in dissociated cultures and were observed both in fixed and dynamically imaged oRG cells.

We are restricted in our ability to monitor primary tissue samples for structural changes over time, so to evaluate long-term changes to cortical development after disruption of mTOR signaling, we utilized pluripotent stem cell (PSC)-derived cortical organoids. Organoids were exposed to the same pharmacological treatments from five weeks of differentiation until collection at 10 weeks (*Figure 2B*). By 10 weeks, HOPX+ oRG cells in cortical organoids are arranged in a rosette-like formation with a rudimentary radial scaffold. There was a significant decrease in the length of the HOPX+ oRG basal processes following either mTOR inhibition or hyperactivation (*Figure 2D*; n = 6 organoids/group from three experiments), similar to the phenotype observed with short term treatments in primary tissue slices and dissociated oRG cells.

We also utilized a strategy of sparse electroporation of progenitors to introduce shRNA constructs that target distinct parts of the mTOR signaling pathway. shRNA against *RPTOR* (*Sarbassov et al., 2005*), part of mTORC1, inactivates the pathway, whereas shRNA against TSC2 (*Vander Haar et al., 2007*), a negative regulator of mTOR, hyperactivates signaling. In primary cortical tissue slices and organoids, *RPTOR* shRNA electroporation resulted in decreased pS6 levels in GFP-expressing electroporated cells, while *TSC2* shRNA increased pS6 levels in primary tissue, supporting the efficacy of the mTOR manipulation (*Figure 3—figure supplement 1C–D*). Electroporated oRG cells were identified by co-expression of GFP and HOPX, and the lengths of GFP+ basal processes were measured. In both culture systems, the oRG basal processes of *RPTOR* shRNA and *TSC2* shRNA electroporated cells were significantly shorter than in controls (primary: n > 12 cells/group from >3 slices/group across three experiments; organoids: n > 27 cells/group from nine organoids across four experiments).

Despite dramatic changes to oRG morphology, manipulation of mTOR signaling did not result in significant changes to the number of HOPX+ oRG cells in organotypic slice cultures and only minimal changes in one mTOR activation condition in organoids (*Figure 2—figure supplement 3*; primary: n > 9 slices/group from eight experiments; organoids: n > 50 sections/group, six organoids/group from three experiments). Notably, modulating mTOR signaling did not result in changes to the cell fate of other progenitor types in primary slice cultures, such as SOX2+ vRGs, TBR2+ IPCs, or neurons destined for the deep or upper layers expressing CTIP2, SATB2 or CUX1 (n > 3 slices/group from >3 independent experiments). In the organoid models, there were modest changes to progenitor and neuron proportions in some mTOR manipulation conditions. However, these results were inconsistent across PSC lines and conditions. Therefore, the organoid model may not be a reliable indicator of mTOR-induced changes in cell number (*Figure 2—figure supplement 4*; n > 34 sections/group, six organoids/group from three experiments). Moreover, the results did not reflect the observations

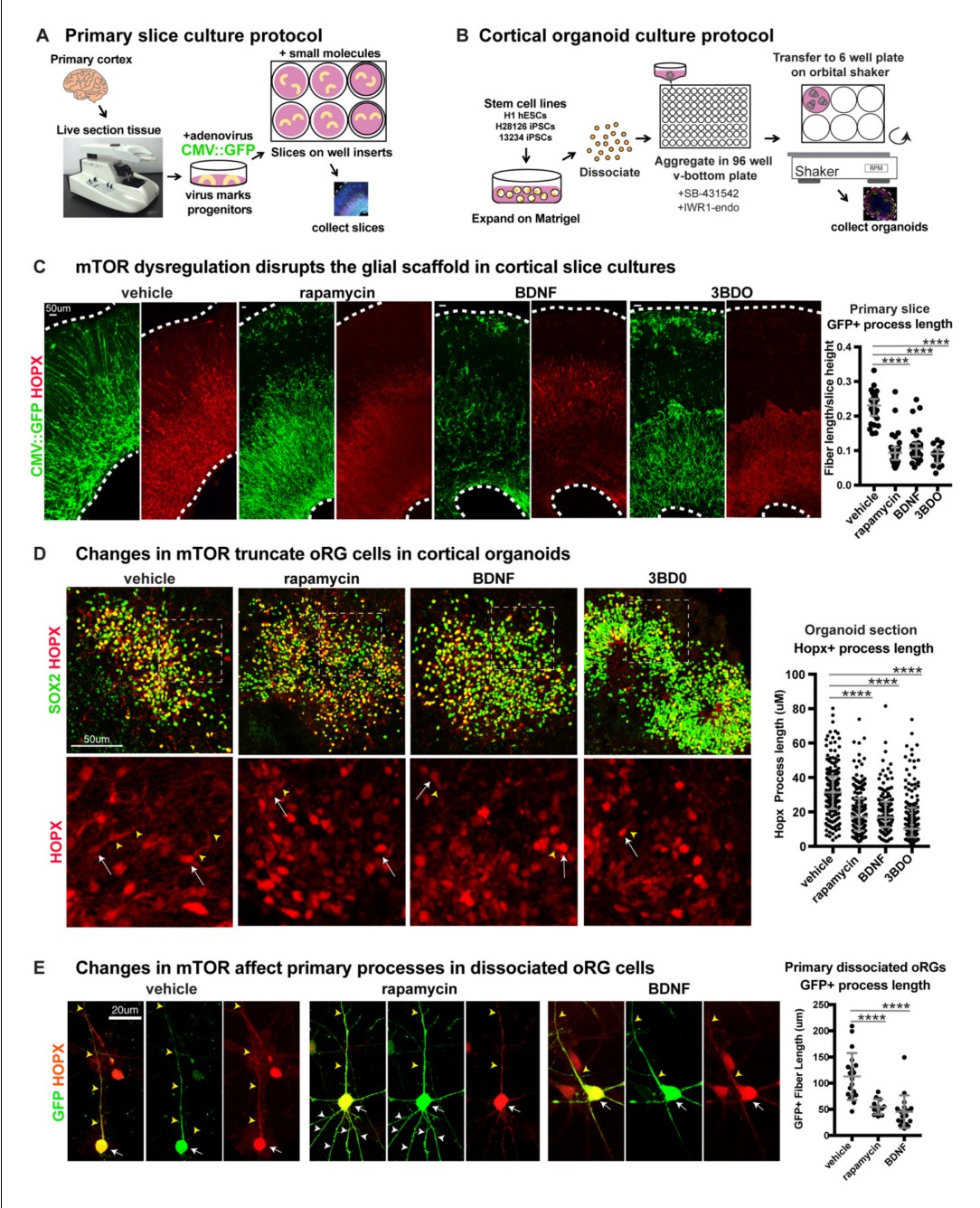

**Figure 2.** Manipulating mTOR signaling disrupts the glial scaffold in primary cortical cultures and organoids. (A) Primary cortical tissue was acutely sectioned and cultured at the air-liquid interface. Small molecules were added the next day and cultured tissue slices were collected six days later. (B) Human pluripotent stem cells were dissociated and aggregated in medium inhibiting Smad and Wnt signaling to promote forebrain induction of cortical organoids. Small molecules were added at five weeks and organoids collected at 10 weeks. (C) Changes to mTOR signaling result in GFP+ HOPX+ glial scaffold disruption in primary cultures. Following mTOR manipulation with rapamycin, 3BD0 and BDNF, the length of GFP+ radial fibers in the oSVZ is significantly reduced (n = 10 vehicle, n = 9 rapamycin, n = 9 BDNF, n = 5 3BDO treated slices from >5 independent experiments; D'Agostino Pearson Normality Test: not normally distributed; Kruskal-Wallis Test: rapamycin: ****p<0.0001, BDNF: ****p<0.0001, ****3BD0: p<0.0001, median with interquartile range shown). (D) Manipulations of mTOR signaling result in HOPX+ oRG fiber truncation and changes to morphology in week 10 cortical organoids. White arrows indicate the cell body and yellow arrowheads illustrate process length. The length of the basal process is significantly smaller in HOPX expressing oRG cells (n = 171 control, n = 156 rapamycin, n = 141 BDNF and n = 162 3BD0 HOPX+ cells each from six organoids/group from three PSC lines; D'Agostino Pearson Normality Test: not normally distributed; Kruskal-Wallis Test: rapamycin: ****p<0.0001, BDNF: ****p<0.0001, ****3BD0: p<0.0001, median with interquartile range shown). (E) Primary dissociated oRG cells treated with mTOR modulators have shorter HOPX+ primary fibers and protrusion of other inappropriate processes. White arrows indicate cell body, yellow arrowheads indicate

*Figure 2 continued on next page*

*Figure 2 continued*

primary process and white arrowheads other processes (n = 19 vehicle, 12 rapamycin and 18 BDNF treated progenitor cells from three independent experiments; D'Agostino Pearson Normality Test: normally distributed; one-way ANOVA: rapamycin: ****p<0.0001, BDNF: ****p<0.0001; error bars represent SD).

The online version of this article includes the following source data and figure supplement(s) for figure 2:

**Source data 1.**

**Figure supplement 1.** Manipulations of mTOR can be monitored through changes to pS6.

**Figure supplement 1—source data 1.**

**Figure supplement 2.** mTOR signaling has functional effects only on oRG progenitor cells.

**Figure supplement 2—source data 1.**

**Figure supplement 3.** Manipulating mTOR signaling does not affect progenitor or neuronal numbers in slice cultures.

**Figure supplement 3—source data 1.**

**Figure supplement 4.** Effects of mTOR manipulations are variable across PSC lines and conditions resulting in little overall effect on progenitor and neuron numbers.

**Figure supplement 4—source data 1.**

**Figure supplement 5.** Changes to mTOR signaling do not affect cell cycle or cell death.

**Figure supplement 5—source data 1.**

from slice culture experiments, a more cytoarchitecturally accurate model of human cortex development (*Bhaduri et al., 2020*). Therefore, we concluded that mTOR signaling has little effect on oRG cell fate. Additionally, mTOR modulation did not alter proliferation, as indicated by BrdU incorporation or presence of the mitotic marker, phospho-histone-H3 (pH-H3), or induce cell death, as indicated by cleaved-caspase 3, in primary slices or organoid cultures (*Figure 2—figure supplement 5*; primary: n > 3 slices/group from five experiments; organoids: n > 19 sections/group four organoids/group from four experiments). Together, our results demonstrate a specific requirement for mTOR signaling to maintain oRG morphology, but not oRG specification or proliferation.

## Manipulation of mTOR signaling reduces oRG migration

oRG cells are characterized by their distinctive MST and migration behaviors. Since mTOR signaling regulates oRG morphology, we queried whether changes to mTOR signaling in the human cortex impacts oRG division and migration. Using sparse electroporation, we assayed oRG migration distance in developing cortical slices. After five days in culture, VZ-labeled electroporated GFP+ cells migrated into the oSVZ (*Figure 3—figure supplement 1E*). We observed that HOPX+ GFP+ oRG cells electroporated with either *RPTOR* or *TSC2* shRNA constructs to cell autonomously inhibit or hyperactivate mTOR respectively, migrated a significantly shorter distance than control oRG cells (*Figure 3B*; n > 27 cells/group from >3 slices/group across three independent experiments). The shorter migration distance after mTOR manipulation suggests not only changes to oRG cell structure, but also potential effects on their migratory or mitotic behavior.

To evaluate a functional role for mTOR signaling in oRG behavior, we dissociated primary human cortical progenitors and labeled them with a GFP-expressing adenovirus. Following treatment with either vehicle or the mTOR inhibitor, rapamycin, oRG migration and mitotic behavior were evaluated using dynamic imaging (*Figure 3C*). oRG cells were identified in dissociated progenitor cultures by their characteristic 'jump and divide' MST behavior, as described previously (Materials and methods; [*Ostrem et al., 2014*]). We observed that modulation of mTOR signaling had no effect on the overall frequency of divisions or the ability of oRG cells to undergo MST (*Figure 2—figure supplement 5C*). However, the translocation distance of the cell body (jump) during MST significantly decreased after mTOR inhibition, suggesting a defect in the motility of oRG cells, as reflected in electroporation studies (*Figure 3D*; n > 8 cells/group from two experiments).

As oRG cells are highly migratory, we also assayed the distance they move prior to division and observed a significant decrease in migration distance as a consequence of mTOR inhibition (*Figure 3E*; n > 8 cells/group from two experiments). Further, in agreement with our observations in cortical slices and organoids, we observed significant morphological changes to oRG cells following mTOR inhibition. The length of the primary fiber was significantly reduced (*Figure 2E*) and we observed extension and retraction of multiple other processes from the oRG cell body that were not observed in vehicle-treated cells (*Figure 3D*, *Figure 2—figure supplement 2E*). The failure to

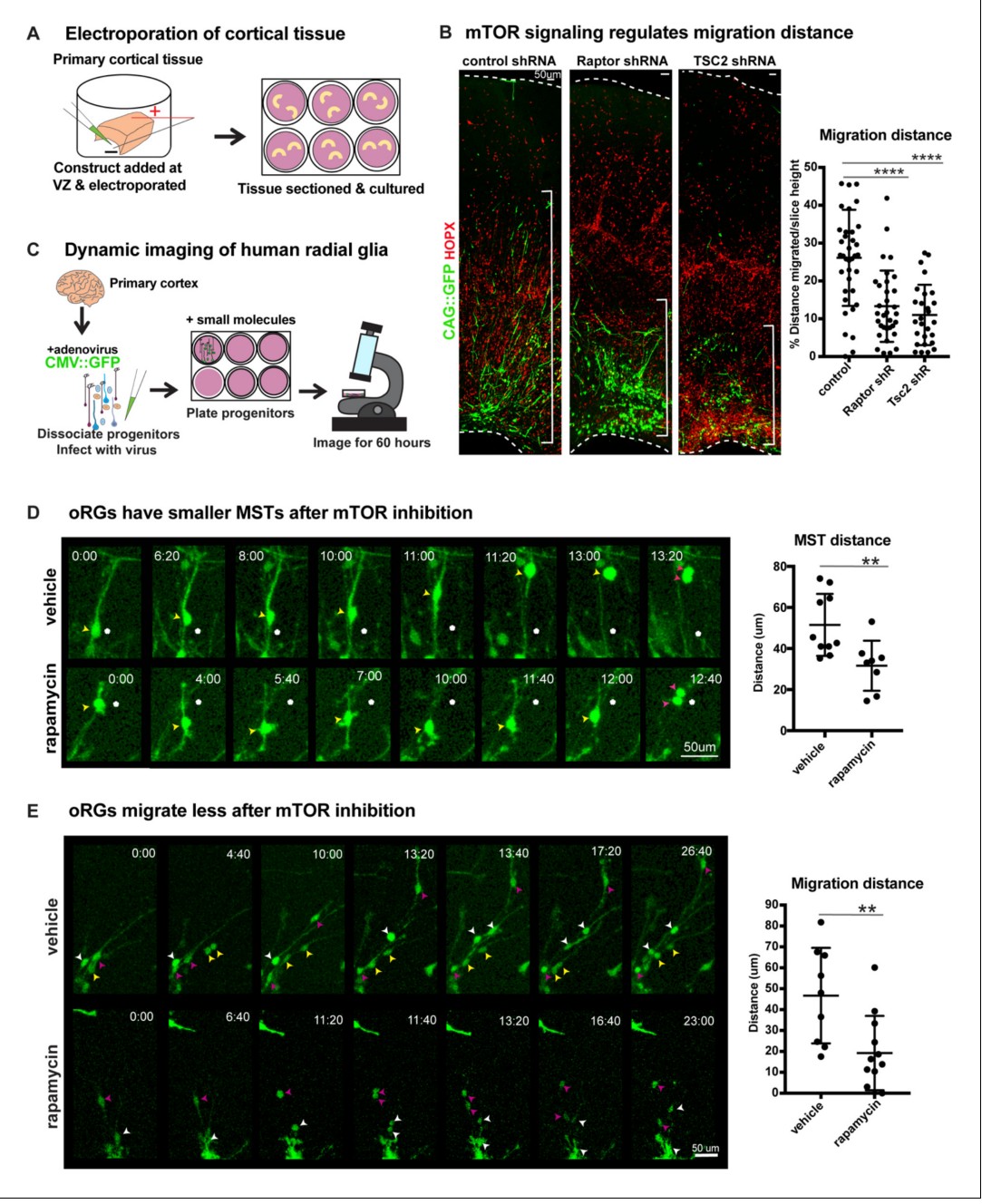

**Figure 3.** Manipulation of mTOR signaling results in migration defects. (**A**) shRNAs were delivered along and electroporated onto the ventricular surface of primary cortical tissue which was then acutely sectioned and cultured for six days prior to collection. (**B**) After electroporation of *RPTOR* or *TSC2* shRNAs, GFP+ HOPX+ oRGs migrate less from the ventricular edge (n = 37 control, n = 33 Raptor and n = 27 TSC2-shRNA electroporated GFP +Hopx+ cells from three independent experiments; D'Agostino Pearson Normality Test: normally distributed; one-way ANOVA with multiple comparisons: Raptor shRNA: ****p<0.0001, TSC2 shRNA: ****p<0.0001, error bars represent SD). The distance of each HOPX+ GFP+ cell away from the VZ edge was measured as indicated by white brackets. (**C**) For dynamic imaging studies, primary cortical tissue was collected, dissociated, infected with a CMV::GFP adenovirus, and plated on glass-bottom 12 well plates. Small molecules were added one day later, two hours before the start of dynamic imaging. (**D**) GFP+ oRG cells undergo division via MST. Yellow arrowheads indicate cell body, white dot indicates initial position of cell body and pink arrowheads indicate two cell bodies after division. After inhibition of mTOR signaling oRG cells have shorter MSTs (n = 10 control and n = 8 rapamycin cells across two independent experiments; D'Agostino Pearson Normality Test: normally distributed; unpaired two-tailed student's t-tests: **p<0.0082, error bars represent SD). (**E**) oRGs migrate less from their original position after mTOR inhibition (n = 9 vehicle treated and n = 12 rapamycin treated cells from two independent experiments; D'Agostino Pearson Normality Test: normally distributed; unpaired two-tailed student's t-tests: **p<0.0058 error bars represent SD). White, yellow, and pink arrowheads indicate cell bodies at starting time-point. Multiple arrowheads of the same color over time indicate daughter cells from the same parent cell.

*Figure 3 continued on next page*

*Figure 3 continued*

The online version of this article includes the following source data and figure supplement(s) for figure 3:

**Source data 1.**
**Figure supplement 1.** shRNAs against mTOR pathway components cell autonomously regulate oRG process length.
**Figure supplement 1—source data 1.**

stabilize the primary fiber after mTOR inhibition points to dysregulation in the cytoskeletal organization of oRG cells, leading to changes in cell morphology and motility.

## Radial fiber truncation following mTOR dysregulation is rescued by CDC42/RAC1 activation

We next sought to explore the mechanism by which mTOR signaling might regulate oRG basal processes and migration behavior. We observed that the levels of filamentous (F) actin in primary dissociated cells, as indicated by phalloidin-rhodamine abundance, significantly decreased after mTOR inhibition or hyperactivation (*Figure 4A*; n > 7 cells/group from three independent experiments). The Rho-GTPases, RHOA, RAC1 and CDC42, are actin regulators and prime candidate molecules to affect radial glial morphology and migration as Rho-ROCK signaling is required for oRG MST behavior (*Hansen et al., 2017*; *Ostrem et al., 2014*) and CDC42 and RAC1 maintain radial glial polarity during mouse cortex development (*Cappello et al., 2006*; *Leone et al., 2010*; *Yokota et al., 2010*). Dissociated primary oRG cells express CDC42, which decreases after treatment with rapamycin or BDNF, suggesting that optimal mTOR signaling maintains total CDC42 levels (*Figure 4C*; n > 3 cells/group from five independent experiments). Downstream effectors of CDC42, Cofilin (CFL1) and Arp2 (ACTR2) are also expressed in oRG cells and decrease after changes to mTOR signaling (*Figure 4C*, *Figure 4—figure supplement 1C*). However, when rapamycin treatment was paired with the GTPase activator, CN04 (*Flatau et al., 1997*; *Lerm et al., 1999*) (Cytoskeleton Inc), expression of CDC42, CFL1, and ACTR2 were rescued (*Figure 4C*).

Changes in mTOR signaling had modest effects on Rho-GTPase activity in cortical tissue overall, potentially due to the other abundant cellular populations in the microdissected cortical tissue that are not regulated by, or that do not respond to, mTOR signaling. However, the GTPase activator, CN04, significantly increased the activity of CDC42 and RAC1, but not RHOA, with CDC42 activity increasing two-fold more than RAC1 (*Figure 4—figure supplement 1A*; n = 3 experiments/protein/ group). Moreover, only CDC42 activity was lowered in slice cultures co-treated with CN04 and rapamycin when compared to cultures treated with CN04 alone, confirming the interaction between mTOR signaling and CDC42 activity. To evaluate the relationship between mTOR signaling and GTPase activity on oRG morphology and migration, we used rapamycin to inactivate mTOR signaling in cortical slice cultures and co-treated the slices with CN04, resulting in a complete rescue of both basal process truncation and oRG migration defects (*Figure 4B*; n > 12 cells/group from >4 slices across three experiments). pS6 expression, however, remained low, indicating that there was no rescue of mTOR activity following CN04 treatment (*Figure 4—figure supplement 1B*; n = 3 experiments). Together, these results suggest that mTOR signaling modulates Rho-GTPase mediated organization of the actin cytoskeleton to regulate the appropriate morphology and migration of human oRG cells.

## Discussion

Our findings demonstrate how mTOR signaling regulates cortical structure and highlights the importance of oRG cells in generating and maintaining radial architecture in the developing human cortex. The mTOR signaling pathway is dysregulated in many neurodevelopmental disorders including autism and Fragile X Syndrome, as well as in cortical malformations such as Focal Cortical Dysplasia and Tuberous Sclerosis. These complex neurodevelopmental disorders arise as a result of accumulated errors throughout development affecting both proliferative cells as well as the migration, morphology, connectivity and plasticity of neurons and glia (*Subramanian et al., 2019*). Discrete short-term and long-term changes to mTOR signaling can therefore create dynamic structural and functional changes in the brain, complicating our understanding of these diseases. A thorough

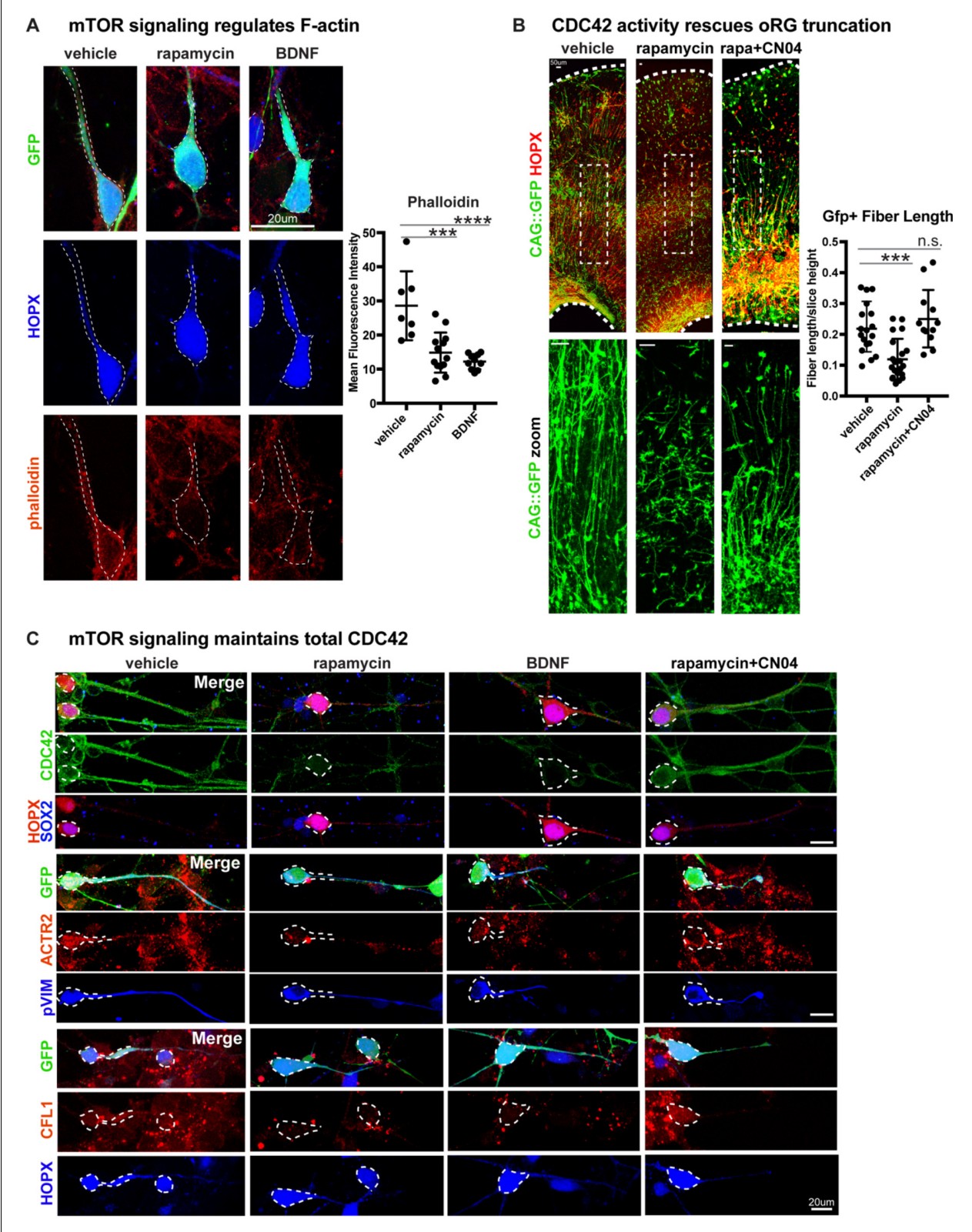

**Figure 4.** Radial fiber truncation from mTOR dysregulation is mediated by actin and rescued by CDC42/RAC1 activation. (**A**) Changes in mTOR signaling decrease the - F-actin in oRG cells as indicated by the reduced intensity of phalloidin-rhodamine staining (n = 7 control, n = 13 rapamycin, n = 10 BDNF cells from three independent experiments; D'Agostino Pearson Normality Test: normally distributed; one-way ANOVA with multiple comparisons: rapamycin: ***p<0.0002, BDNF: ****p<0.0001, error bars represent SD). (**B**) mTOR loss disrupts oRG fiber length, which can be rescued

*Figure 4 continued on next page*

*Figure 4 continued*

by Rho-GTPase activation using CN04 (slice culture GFP+ fiber length: n = 18 control, n = 21 rapamycin and n = 12 CN04+rapamycin-treated cells from n > 4 slices/group across three independent experiments; D'Agostino Pearson Normality Test: normally distributed; one-way ANOVA with multiple comparisons: rapamycin: ***p<0.0007, rapamycin+CN04: p=0.5097, error bars represent SD). (C) The Rho-GTPase CDC42 is expressed in primary dissociated oRG cells, but protein abundance decreases after manipulation of mTOR signaling. Mediators of CDC42 activity, including ACTR2 and CFL1, are also present in dissociated oRG cells. Levels of ACTR2 and CFL1 protein decrease after mTOR manipulation, but can be restored after activation of Rho-GTPases with CN04 (five independent experiments).

The online version of this article includes the following source data and figure supplement(s) for figure 4:

**Source data 1.**
**Figure supplement 1.** mTOR signaling modulation changes CDC42 and RAC1 activity.
**Figure supplement 1—source data 1.**

characterization of these different effects of mTOR dysregulation in human cortical development is therefore crucial to the creation of better disease models and therapies for multiple neurodevelopmental disorders. In this study, we document the role of mTOR signaling in oRG cells during the neurogenic period of human cortical development. We observe specific activation and function of mTORC1 signaling in human oRG cells, but not other neural progenitor subtypes, during peak neurogenesis. Further, we identify a unique role for the mTOR signaling pathway in regulating oRG morphology and migration. The polar morphology of oRG cells, with an elongated basal process, not only provides a mechanism for the dynamic division and migratory behavior that promotes progenitor expansion (*Betizeau et al., 2013*; *Gertz and Kriegstein, 2015*; *Kalebic et al., 2019*; *LaMonica et al., 2013*), but also creates a scaffold for the migration of cortical neurons. Our results show that mTOR signaling in oRG cells is tightly regulated to maintain cell morphology. Dysregulation of mTOR signaling results in truncation of the basal process and oRG migration defects. Furthermore, mTOR signaling activity is transduced by CDC42/RAC1-mediated regulation of the actin cytoskeleton, and defects after mTOR loss can be rescued by CDC42/RAC1 activation (*Figure 5*). By regulating the morphology of oRG cells and modulating their migratory behavior, mTOR signaling plays a central role in the development of the human cerebral cortex.

In most cells, mTORC1 regulates protein translation by controlling ribosome assembly and translation initiation, while the mTOR complex 2 (mTORC2) functions upstream of Rho GTPases to regulate the actin cytoskeleton (*Jacinto et al., 2004*; *Saxton and Sabatini, 2017*). Given the cytoskeletal

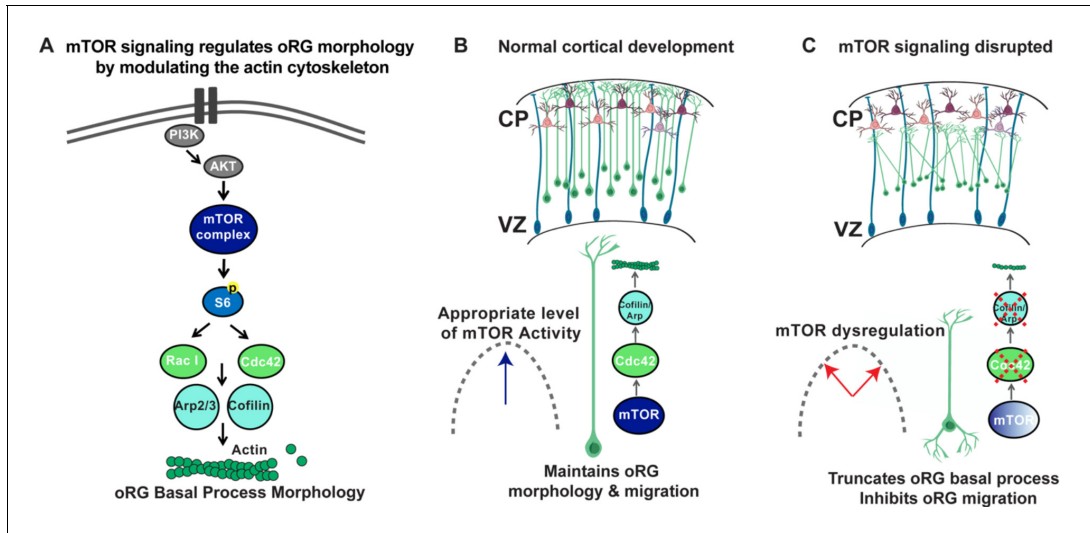

**Figure 5.** Model of mTOR regulation of oRG morphology. (A) Simplified model of mTOR signaling pathway activating the GTPases, CDC42 and RAC1 to regulate actin dynamics. (B) During normal development appropriate levels of mTOR signaling regulate oRG morphology, basal process length and migration behavior through activation of CDC42 and its targets, CFL1 and ACTR2. (C) When mTOR signaling is dysregulated the basal process is truncated and oRG cells have decreased levels of F-actin, CDC42, CFL1, and ACTR2. The oRG cells are mispositioned, the basal process does not reach the pial surface, and the glial scaffold is disrupted.

effects of mTOR signaling in oRG cells, the mTORC2 pathway could be the transducer of mTOR signaling in oRG cells. However, mTORC2 signaling is RPTOR-independent and less sensitive to rapamycin (*Jacinto et al., 2004*). In oRG cells, both rapamycin treatment and RPTOR shRNA have clear effects on the cytoskeleton, similar to the downstream effects of mTORC2. Prolonged rapamycin exposure may affect the assembly of mTORC2 complexes (*Sarbassov et al., 2006*), but the cytoskeletal effects of RPTOR shRNA remain puzzling. Instead, it is possible that mTORC1 controls translation initiation and local mRNA transport (*Pilaz et al., 2016*; *Pilaz and Silver, 2017*) into the basal processes of oRG cells. Recent studies have identified direct regulation of translation of the human *ß-Actin* mRNA by mTORC1 and shown that this regulation may be cell-type dependent (*Eliseeva et al., 2019*). mTORC1 may therefore, indirectly affect the ability of oRGs to generate cytoskeletal proteins needed for basal process elongation and maintenance by regulating protein synthesis. Both mTOR complexes, mTORC1 and C2, could thus control the cytoskeletal organization of the basal process in oRG cells, suggesting that the morphology and cell biology of the basal process may represent a crucial regulatory hub for cortical expansion.

In migrating neurons in the mammalian cortex, actin filaments interact with myosin motor proteins to control aspects of cell motility via actomyosin regulation (*Solecki et al., 2009*). We observed a particular division of labor between the actin cytoskeleton and non-muscle myosin II in human oRGs. Previous studies have demonstrated the importance of the molecular motor, non-muscle myosin-II, along with active Rho-ROCK signaling for the generation of MSTs - the characteristic 'jump and divide' mitotic behavior unique to oRGs (*Ostrem et al., 2014*). mTOR signaling appears to affect the migration of oRGs prior to and during mitosis, without actually affecting the MST itself. mTOR signaling alters actin and its regulators in oRG cells but does not affect RHO A activity. Thus, the components of the actomyosin system appear to have complementary roles in regulating different aspects of oRG movement, with Rho-mediated non-muscle myosins regulating MSTs while migration itself is controlled by actin-modulating GTPases, such as CDC42 and RAC1. Effects of CDC42/RAC1 signaling on the morphology, polarity and proliferation of ventricular radial glia (vRG) have been reported in the mouse cortex (*Cappello et al., 2006*; *Leone et al., 2010*; *Yokota et al., 2010*), suggesting that the specific activity of mTOR signaling in human oRG cells may control their morphology and migration through conserved molecular pathways.

The reasons behind the bimodal effects of mTOR signaling in human oRG cells remain unclear. Our observations of rapid and sustained changes to cell morphology following mTOR modulation in oRG cells suggest that these effects occur at the protein level, either by phosphorylation of key cytoskeletal mediators or due to the translational regulation of cytoskeletal mRNAs. mTORC1 and C2 act in large multi-protein complexes to control both aspects of protein regulation. As a result, optimum levels of individual proteins are needed for the proper assembly of functional complexes (*Ardestani et al., 2018*; *Eliseeva et al., 2019*; *Varusai and Nguyen, 2018*). It is probable that too much or too little of any of the individual proteins will result in abnormal functional protein assemblies and disrupted cytoskeleton in oRG cells. Similar observations detailing the bimodal effects of mTOR signaling on cellular phenomena have been made in other neurological systems. For example, mTOR signaling appears to function within an optimal activity range to regulate synaptic plasticity. Alterations leading to excessive increase or decrease in mTOR signaling have detrimental effects on both learning and long-term memory retention (*Oddo, 2012*). Interestingly, the effects of mTOR on learning and memory likely occur via local translational control of protein synthesis in dendrites, which in turn regulate the structural plasticity of synapses (*Ehninger et al., 2008*; *Oddo, 2012*; *Puighermanal et al., 2009*; *Tang et al., 2002*). Similar translational control may operate in the basal processes of oRG cells, regulating the abundance of cytoskeletal proteins by maintaining optimal levels of mTOR signaling.

Future experiments will more deeply probe the molecular pathways by which mTOR signaling mediates its effects on the cytoskeleton of human oRG cells. Exploring the complexity of mTOR pathway interactions in the developing human cortex is particularly challenging given the limited availability of primary tissue. Cortical organoids, therefore, present an attractive option for future exploration as they are stem cell-derived, can be maintained long term, and are well-suited for molecular and genetic manipulations. While cortical organoids do not currently replicate all aspects of developing cortical organization and metabolism (*Amiri et al., 2018*; *Bhaduri et al., 2020*; *Camp et al., 2015*; *Pollen et al., 2019*), many molecular and cellular developmental programs are conserved (*Bershteyn et al., 2017*; *Lancaster et al., 2013*; *Quadrato et al., 2017*; *Sloan et al.,*

*2017*; *Subramanian et al., 2017*; *Velasco et al., 2019*). In this study, we demonstrate that the mTOR signaling pathway has similar expression and functional effects in both primary human tissue and organoids. Future studies using human in vitro models will better resolve which mTOR complex proteins control cytoskeletal GTPases and clarify the relationship between actin regulation and oRG morphology, insights that may shed light on the mechanisms of human neurodevelopmental disease.

# Materials and methods

## Key resources table

| Reagent type (species) or resource | Designation | Source or reference | Identifiers | Additional information |
|---|---|---|---|---|
| Cell line (*Homo sapiens*) | H1/WA01 embryonic stem cell line | WiCell | RRID:CVCL_9771 | Male |
| Cell line (*Homo sapiens*) | H28126 induced pluripotent stem cell line | *Pollen et al., 2019* | | Male |
| Cell line (*Homo sapiens*) | 13234 induced pluripotent stem cell line | *Bhaduri et al., 2020* | RRID:CVCL_0G84 | Female |
| Transfected construct (*Homo sapiens*) | CMV::GFP Adenovirus | Vector Biolabs, 1060 | | 1:200 dilution |
| Transfected construct (*Aequorea victoria*) | pCAG-EGFP | *Subramanian et al., 2011* | | Electroporation |
| Transfected construct (*Homo sapiens*) | Scramble shRNA | Addgene plasmid # 1864 | RRID:Addgene_1864 | Electroporation (1 µg/ul) |
| Transfected construct (*Homo sapiens*) | Raptor_1 shRNA | Addgene plasmid # 1857 | RRID:Addgene_1857 | Electroporation (1 µg/ul) |
| Transfected construct (*Homo sapiens*) | pLKO.1-TSC2 | Addgene plasmid # 15478 | RRID:Addgene_15478 | Electroporation (1 µg/ul) |
| Biological sample (*Homo sapiens*) | Primary Cortex Tissue Samples | UCSF Gamete, Embryo and Stem Cell Research Committee (GESCR) approval | | GW16-19 |
| Antibody | anti-Sox2 (Mouse monoclonal) | Santa Cruz | RRID:AB_10842165; Cat# sc-365823 | (1:500) |
| Antibody | anti-Hopx (Mouse monoclonal) | Santa Cruz | RRID:AB_2687966; Cat# sc-398703 | (1:250) |
| Antibody | anti-Cofilin (Mouse monoclonal) | Santa Cruz | RRID:AB_11150468; Cat# sc-376476 | (1:100) |
| Antibody | anti-Cdc42 (Mouse monoclonal) | Santa Cruz | RRID:AB_627233; Cat#, sc-8401 | (1:200) |
| Antibody | anti-Cux1 (Mouse monoclonal) | Abcam | RRID:AB_941209; Cat#AB54583 | (1:500) |
| Antibody | anti-pVim (Mouse monoclonal) | MBL International | RRID:AB_592969; Cat# D095-3 | (1:500) |
| Antibody | anti-pHistone H3 (Mouse monoclonal) | Abcam | RRID:AB_443110; Cat# ab14955 | (1:500) |
| Antibody | anti-Cryab (Mouse monoclonal) | Abcam | RRID:AB_300400; Cat# ab13496 | (1:500) |
| Antibody | anti-Hopx (Rabbit polyclonal) | Proteintech | RRID:AB_10693525; Cat# 11419–1-AP | (1:500) |
| Antibody | anti-pS6 (Rabbit polyclonal) | Cell Signaling | RRID:AB_331679; Cat# 2211S | (1:500) |

*Continued on next page*

*Continued*

| Reagent type (species) or resource | Designation | Source or reference | Identifiers | Additional information |
|---|---|---|---|---|
| Antibody | anti-p4EBP1 (Rabbit monoclonal) | Cell Signaling Technology | RRID:AB_560835; Cat# 2855S | (1:500) |
| Antibody | anti-DEPDC5 (Rabbit polyclonal) | Sigma | RRID:AB_2682655; Cat# HPA054969 | (1:500) |
| Antibody | anti-Arp2 (Rabbit polyclonal) | Proteintech | RRID:AB_2221854; Cat#10922–1-AP | (1:100) |
| Antibody | anti-Cleaved Caspase-3 (Rabbit polyclonal) | Cell Signaling Technology | RRID:AB_2341188; Cat#9661S | (1:200) |
| Antibody | anti-Ctip2 (Rat monoclonal) | Abcam | RRID:AB_2064130; Cat# ab18465 | (1:500) |
| Antibody | anti-BrdU (Rat monoclonal) | Abcam | RRID:AB_305426; Cat# ab6326 | (1:500) |
| Antibody | anti-Eomes (Sheep polyclonal) | R and D | RRID:AB_10569705; Cat# AF6166 | (1:200) |
| Antibody | anti-Sox2 (Goat polyclonal) | Santa Cruz | RRID:AB_2286684; Cat# sc-17320 | (1:250) |
| Antibody | anti-NeuN (Guinea pig polyclonal) | Millipore | RRID:AB_11205592; Cat# ABN90 | (1:500) |
| Antibody | anti-Gfp (Chicken polyclonal) | Aves | RRID:AB_10000240; Cat#, GFP-1020 | (1:500) |
| Commercial assay or kit | G-LISA Activation Assay | Cytoskeleton, Inc, | catalog# BK124, BK127, BK128 | |
| Peptide, recombinant protein | BDNF | Peprotech | Cat# 450–02 | 100 ng/mL |
| Chemical compound, drug | Rapamycin | Sigma | Cat# R8781-200UL | 250 nM |
| Chemical compound, drug | 3BD0 | Sigma | Cat# SML1687 | 1 µM |
| Chemical compound, drug | CN04 | Cytoskeleton inc, | Cat# CN04 | 500 ng/mL |
| Software, algorithm | Imaris | Oxford Instruments | Imaris, RRID:SCR_007370 V9.5 | |

## Lead contact and materials availability

Questions and requests for resources should be directed to the lead contacts, Arnold Kriegstein, at Arnold.Kriegstein@ucsf.edu and Lakshmi Subramanian at lakshmi.subramanian.me@gmail.com. No unique materials or reagents were produced from this study.

## Experimental model and subject details

### Pluripotent Stem Cell lines

H1/WA01 human embryonic stem cell line
H28126 human induced pluripotent stem cell line
13234 human induced pluripotent stem cell line

### Primary human cortex tissue

All primary tissues were obtained and processed as approved by UCSF Gamete, Embryo and Stem Cell Research Committee (GESCR) approval 10–05113. Tissue was collected with patient consent for research and in strict observance of legal and institutional ethical regulations. All samples were de-identified and no sex information is known.

## Method details

### PSC expansion culture

Human induced pluripotent stem cell lines, H28126 and 13234 (*Burrows et al., 2016*; *Pollen et al., 2019*), and the embryonic stem cells line, H1 (WiCell), were expanded on growth factor-reduced Matrigel (BD)-coated six well plates. Stem cells were thawed in StemFlex Pro Media (Gibco) containing 10 µM Rock inhibitor Y-27632. Medium was changed every other day and lines passaged when colonies reached about 70% confluency. Stem cells were passaged using PBS-EDTA and residual cells manually lifted with cell lifters (Fisher). All lines used for this study were between passage 25–40.

### PSC line authentication

This work uses the following human stem cell lines: WA01/H1, 13234 and H28126. Line sources: WA01: WiCell, 13234: Conklin lab (Gladstone Institute), H28126: Gilad lab (University of Chicago). All of these lines have been authenticated in previous studies (*Bhaduri et al., 2020*). All stem cell lines were karyotyped and validated for pluripotency, prior to receipt. Every 10 passages, stem cells are tested for karyotypic abnormalities and validated for pluripotency markers Sox2, Nanog, and Oct4. All cell lines tested negative for mycoplasma.

### Cortical organoid differentiation protocol

Cortical organoids were cultured using a forebrain directed differentiation protocol (*Kadoshima et al., 2013*; *Pollen et al., 2019*). Briefly, PSC lines were expanded and dissociated to single cells using accutase. After dissociation, cells were reconstituted in neural induction medium at a density of 10,000 cells per well in 96 well v-bottom low adhesion plates. GMEM-based neural induction medium includes 20% Knockout Serum Replacer (KSR), 1X non-essential amino acids, 0.11 mg/mL Sodium Pyruvate, 1X Penicillin-Streptomycin, 0.1 mM Beta Mercaptoethanol, 5uM SB431542 and 3 µM IWR1-endo. Medium was supplemented with 20 µM Rock inhibitor Y-27632 for the first 6 days. After 18 days organoids were transferred from 96 to six well low adhesion plates and moved to an orbital shaker rotating at 90 rpm and changed to DMEM/F12-based medium containing 1X Glutamax, 1X N2, 1X CD Lipid Concentrate and 1X Penicillin-Streptomycin. At 35 days, organoids were moved into DMEM/F12-based medium containing 10% FBS, 5 µg/mL Heparin, 1X N2, 1X CD Lipid Concentrate and 0.5% Matrigel. At 70 days medium was additionally supplemented with 1X B27 and Matrigel concentration increased to 1%. Throughout culture duration organoids were fed every other day. Organoids were collected for immunohistochemistry at weeks 8 and 10 of differentiation. mTOR experimental groups were treated with 250 nM rapamycin, 100 ng/mL BDNF or 1 µM 3BDO.

### Organotypic slice culture

Primary cortical tissue was maintained in artificial cerebrospinal fluid(125 mm NaCl, 2.5 mm KCl, 1 mm MgCl2, 2 mm CaCl$_2$, 1.25 mm NaH2PO$_4$, 25 mm NaHCO$_3$, 25 mm d-(+)-glucose) bubbled with 95% O$_2$/5% CO$_2$ until embedded in a 3% low melt agarose gel. Embedded tissue was acute sectioned at 300 µM using a vibratome (Leica) and plated on Millicell (Millipore) inserts in a six well tissue culture plate. Slices were cultured at the air liquid interface in medium containing 32% Hanks BSS, 60% BME, 5% FBS, 1% glucose, 1% N2 and 1% Penicillin-Streptomycin-Glutamine. Slices were maintained for 7 days in culture at 37°C and medium changed every third day. mTOR experimental groups were treated with 250 nM rapamycin, 100 ng/mL BDNF, 1 µM 3BDO or 500 ng/mL CN04 (Cytoskeleton inc).

### Dissociated cell culture and live imaging

Primary human cortical tissue was dissociated with Papain (Worthington)-containing DNase. Samples were microdissected to enrich for progenitors, by removing the cortical plate. Tissue was coarsely chopped and submerged in 5 mL of Papain and incubated at 37°C for 15mins. Samples were inverted three times and continued incubating at 37C for another 15 mins. Samples were then triturated by manual pipetting approximately ten times. Dissociated cells were spun down at 300 g for 5mins and Papain removed. Cells were reconstituted in DMEM/F12-based medium containing 1X Glutamax, 1X B27, 1XN27, 1X sodium pyruvate and 1x penicillin-streptomycin. Cells were plated at

1 million/mL in 12 well matrigel-coated glass-bottom tissue culture plates or eight chamber glass slides. For timelapse imaging, cultures were then transferred to an inverted Leica TCS SP5 with an on-stage incubator streaming 5% $CO_2$, 8% $O_2$, and balanced $N_2$ into the chamber. Primary dissociated cells were imaged for GFP using a 10x air objective at 20 min intervals for up to 3 days with repositioning of the z-stacks every 10–12 hr.

## Electroporation
### Organoids
*RPTOR shRNA*, *TSC2 shRNA* (Addgene) and *CAG-GFP* plasmids were electroporated at 1 µg/ul using a BTX electroporator set at 5V for 5 milliseconds for three pulses. Organoids were bathed in 200 µl media containing plasmid for 10mins. Then, an electric current was passed through the cuvette. Organoids were maintained for 4–10 days in culture prior to processing.

### Primary cortical slices
A reporter plasmid expressing GFP under the CAG promoter (pCAG-EGFP at 1 µg/ul concentration) was mixed with the shRNA constructs in a 1:3 ratio to ensure that all GFP positive cells were co-electroporated with the shRNA. The plasmid mixture was applied only to the ventricular surface of thick cortical tissue slices (thickness >2 cm) using a finely beveled glass needle. The slices were electroporated with paddle electrodes where the negative paddle was positioned along the ventricular surface and the positive paddle was positioned over the pial surface. Electric current was applied using a BTX electroporator for 50 ms at 75V. five pulses were applied with an interval of 950 ms between pulses. The electroporated slices were allowed to recover for 10 to 15 min in cold, oxygenated ACSF, acute sectioned and cultured organotypically for 6 days as described earlier.

### Constructs
*pCAG-EGFP* (*Subramanian et al., 2011*) was used as a reporter plasmid. Scramble shRNA was used for control electroporations and was a gift from David Sabatini (Addgene plasmid # 1864; http://n2t.net/addgene:1864; RRID:Addgene_1864). Raptor_1 shRNA was a gift from David Sabatini (Addgene plasmid # 1857; http://n2t.net/addgene:1857; RRID:Addgene_1857). pLKO.1-TSC2 was a gift from Do-Hyung Kim (Addgene plasmid # 15478; http://n2t.net/addgene:15478; RRID:Addgene_15478.).

## Immunohistochemistry
Cortical organoids were collected, fixed in 4% PFA for 45mins, washed with 1xPBS and submerged in 30% sucrose in 1xPBS until saturated. Organoids were embedded in cryomolds containing 50% O.C.T. (Tissue-tek) and 50% of 30% sucrose in 1xPBS and frozen at −80℃. Organoids were sectioned at 16 µM onto glass slides. Antigen-retrieval was performed on tissue sections using a citrate-based antigen retrieval solution at 100x (Vector Labs) which was boiled and tissue submerged in solution for 20mins. After antigen retrieval, slides were briefly washed with PBS and blocked with PBS containing 5% donkey serum, 2% gelatin and 0.1% Triton X-100 for 30mins. Primary antibodies were incubated in blocking buffer on slides at 4C overnight, washed with PBS containing 0.1% Triton X-100 three times and then incubated with AlexaFluor secondary (Thermo Fisher) antibodies at room temperature for 2 hr.

Organotypic slice cultures were fixed for 2 hr in 4% PFA and washed with 1xPBS overnight. Slices were subjected to boiling citrate-based antigen retrieval solution (Vector Labs) for 20 min and permeabilized and blocked in blocking buffer (PBS plus 0.1% Triton X-100, 10% donkey serum, and 0.2% gelatin) for 1 hr at room temperature. Primary antibodies were diluted in blocking buffer and applied to slices for 36 hr at 4℃. Slices were washed with PBS plus 0.5% Triton X-100 and then incubated in AlexaFluor secondary antibodies (Thermo Fisher and Jackson Labs) diluted in blocking buffer at 4℃ overnight. Images were acquired on a Leica TCS SP5 X laser confocal microscope.

### Primary antibodies include
Mouse: Sox2 (Santa Cruz, sc-365823, 1:500), Hopx (Santa Cruz, sc-398703, 1:250), Cofilin (Santa Cruz, sc-376476, 1:100), Cdc42 (Santa Cruz, sc-8401, 1:200) Cux1 (Abcam, AB54583, 1:500), pVim (Fisher, 5045934, 1:500), pHistone H3 (abcam, ab14955, 1:500), Cryab (ab13496, 1:500), Rabbit: Hopx (Proteintech, 11419–1-AP, 1:500), pS6 (Cell Signaling, 2211S, 1:500), p4EBP1 (Cell Signaling

Technology, 2855S, 1:500), DEPDC5 (Sigma, HPA054969, 1:500), Arp2 (Proteintech, 10922–1-AP, 1:100), Cleaved Caspase-3 (9661S, 1:200). Rat: Ctip2 (Abcam, ab18465, 1:500), BrdU (Abcam, ab6326, 1:500). Sheep: Eomes (R and D, AF6166, 1:200), Goat: Sox2 (Santa Cruz, sc-17320, 1:250), Guinea pig: NeuN (Millipore, ABN90, 1:500), Chicken: Gfp (Aves, GFP-1020, 1:500).

## G-LISA activation assay

Activation of Cdc42, RhoA, and RAC1 were performed through the use of an ELISA-based Rho-GTPase activation assay kit (Cytoskeleton, Inc, catalog# BK124, BK127, BK128). Briefly, protein was isolated from the SVZ of microdissected slice cultures after treatment with vehicle, rapamycin, BDNF, CN04 or CN04+rapamycin. Microdissected tissue pieces were triturated with a fine gauge needle in lysis buffer containing a protease inhibitor on ice. Protein concentration was calculated on a cuvette spectrometer (600 nM) using Precision Red Reagent (part #GL50). Concentration was normalized across samples by addition of proportionate volumes of lysis buffer. Normalized protein lysates were then applied to equilibrated Cytoskeleton Inc microplate wells and incubated. Plates were washed in between each of the subsequent steps: incubation with antigen presenting buffer, primary antibody, HRP secondary antibody, HRP detection solution and HRP stop solution. Colorimetric changes compared to negative control were detected using a microplate spectrometer (490 nM). Differences across groups were normalized to control and averaged across experiments.

## Quantification and statistical analysis

### Primary basal fiber length quantification

To quantify the length of individual radial fibers, clearly visible GFP+ radial glia were selected. For all conditions, we identified the GFP+ cell in the oSVZ with the longest basal process, along with at least two additional representative cells for a total of three GFP+ processes per slice culture. The length of the basal process in these cells was measured between its point of origin on the basal side of the cell and its end point. As these were z-stacks of 300 μM tissue slices, the point where the basal process emerged from the cell body on the basal side was found in the appropriate z-plane and marked. The length of the process was traced through the depth of the z-stack and the endpoint marked using the polygon trace tool in Imaris software. The same tool was used to measure the length of the traced process. At least 3 cells/slice image were measured in μMs. The total height of the slice was then determined from the ventricular zone edge to the edge of the cortical plate. Since slice culture size is variable across experiments the length of each radial fiber was calculated as a percent of the total slice height. In both pharmacological and genetic studies, the GFP length of the longest process in all groups, which was also always the primary (basally-oriented) process, was measured.

### Live imaging data analysis

Maximum intensity projections of the collected image stacks (5–10 μm step size) were compiled, generated into movies, and analysed using Imaris software. Maximum intensity projections of individual mitotic cells were examined before, during and after cytokinesis. The fiber was measured from the point of origin at the cell body to the tip of the longest primary branch. oRG cells were identified in dissociated progenitor cultures by their characteristic MST behavior, as described previously (Ostrem et al., 2014). MST was defined as a translocation of greater than or equal to one cell diameter of the soma along the primary process, with a velocity of greater than or equal to 20 μm/hr, coinciding with cell rounding, and ending either in immediate cytokinesis or in a prolonged, rounded state. Migration distance was calculated for mitotic cells that showed MSTs by retrospectively measuring the distance traveled by the cell during 20 imaging timepoints (approximately 16 hr) prior to mitosis.

### Statistical analyses

All experiments, which contained more than two conditions per experiment, were evaluated for normality with the D'Agostino-Pearson omnibus test. If normally distributed, one-way ANOVA with multiple comparisons was performed. If not normally distributed, Kruskall-Wallis Test was performed. For experiments containing only two conditions an unpaired two-tailed student's t-test was used. Within an experiment, the control group was compared to each experimental group (rapamycin,

BDNF, CN04 etc) and significance determined. Significance was assigned when p<0.05. For all groups: *p<0.05, **p<0.01, ***p<0.001, ****p<0.0001.

### Sample sizes

No sample size calculations were performed prior to experiments. Due to limited availability of developing human cortical tissue, we replicated experiments as many times as possible during the timeframe of the study. All experiments included samples from at least two individuals, although most were performed with >3 individuals, and multiple technical replicates within each of those independent experiments. As organoids are a more readily available and renewable resource, we utilized more than three organoids for each experiment. All experiments also contained organoids derived from three different iPSC lines and several experimental replicates were collected from each line.

### Pre-established exclusion criteria

Slice, cell or organoid cultures with bacterial or fungal contamination were excluded. Slice, cell, or organoid cultures that did not show appropriate pharmacological responses to mTOR modulation (as determined by pS6 staining) were excluded from the analysis.

### Replication

All organoid and acute cortical pharmacological experiments (slice cultures and dissociated cells) were repeated at least three times with independent samples to ensure reproducibility. Live imaging and primary electroporation experiments required large intact pieces of developing human cortex that are only rarely available. Some of these experiments were repeated two times using tissue from two different individuals. For organoid experiments, three different pluripotent stem cell lines were used for differentiation. All attempts at replication were successful.

### Randomization

In each experiment, the same tissue samples were sectioned and slices were evenly distributed across experimental conditions. Similar criteria were used for dissociated cell cultures. For organoid experiments, organoids from a given line and differentiation batch were distributed across a six well plate and treated pharmacologically, with each well being a distinct group.

## Acknowledgements

The authors thank Qiuli Bi, Shaohui Wang, Diane Jung, Mohammed Mostajo-Radji, Melanie Bedolli, William Walantus, and members of the ARK laboratory for providing resources, technical help and useful discussions. We thank Diane Barber, Aparna Bhaduri, Stephen Floor, Tom Nowakowski and Alex Pollen for comments on the manuscript. This study was supported by NIH award U01MH114825 and 3R35NS097305 to ARK, the California Institute for Regenerative Medicine (CIRM) through the CIRM Center of Excellence in Stem Cell Genomics (GC1R-06673-C to ARK), the Cure Foundation Taking Flight Award and the NARSAD Young Investigator Award to LS.

## Additional information

### Funding

| Funder | Grant reference number | Author |
| --- | --- | --- |
| National Institutes of Health | U01MH114825 | Arnold Kriegstein |
| National Institutes of Health | 3R35NS097305 | Arnold Kriegstein |
| Brain and Behavior Research Foundation | GRANT_NUMBER: 27729 | Lakshmi Subramanian |
| Citizens United for Research in Epilepsy | P0530360 | Lakshmi Subramanian |
| California Institute of Regenerative Medicine | GC1R-06673-C | Arnold Kriegstein |

The funders had no role in study design, data collection and interpretation, or the decision to submit the work for publication.

## Author contributions

Madeline G Andrews, Conceptualization, Data curation, Formal analysis, Validation, Investigation, Visualization, Methodology, Writing - original draft, Project administration, Writing - review and editing; Lakshmi Subramanian, Conceptualization, Data curation, Formal analysis, Funding acquisition, Validation, Investigation, Visualization, Methodology, Writing - original draft, Project administration, Writing - review and editing; Arnold R Kriegstein, Conceptualization, Resources, Supervision, Funding acquisition, Project administration, Writing - review and editing

## Author ORCIDs

Madeline G Andrews (iD) https://orcid.org/0000-0002-5154-5081
Lakshmi Subramanian (iD) https://orcid.org/0000-0002-3504-4455
Arnold R Kriegstein (iD) https://orcid.org/0000-0001-5742-2990

## Ethics

Human subjects: All primary tissue was obtained and processed as approved by UCSF Gamete, Embryo and Stem Cell Research Committee (GESCR) approval 10-05113. Tissue was collected with patient consent for research and in strict observance of legal and institutional ethical regulations. All samples were de-identified and no sex information is known.

## Decision letter and Author response

Decision letter https://doi.org/10.7554/eLife.58737.sa1
Author response https://doi.org/10.7554/eLife.58737.sa2

# Additional files

## Supplementary files

• Transparent reporting form

## Data availability

All data generated or analyzed during this study are included in the manuscript and supporting files. Source data files are provided all Figures and Figure supplements.

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
