## [Decision Letter]

**Acceptance summary:**

The manuscript describes the results of elegantly performed experiments to understand the functional role of mTOR signaling in human cortical development. Understanding the contribution of mTOR in human cortical development is important, as mTOR signaling is impaired in many neurodevelopmental disorders. The work provides valuable insights into the impact of mTOR signaling in regulating the morphology and behavior of outer radial glia (oRG) in the developing human cortex.

**Decision letter after peer review:**

Thank you for submitting your article "mTOR signaling regulates the morphology and migration of outer radial glia in developing human cortex" for consideration by *eLife*. Your article has been reviewed by three peer reviewers, including Anita Bhattacharyya as the Reviewing Editor and Reviewer #1, and the evaluation has been overseen by Marianne Bronner as the Senior Editor.

The reviewers have discussed the reviews with one another and the Reviewing Editor has drafted this decision to help you prepare a revised submission.

Summary:

The reviewers all agree that the manuscript is elegantly performed and provides an important contribution by providing valuable insights into the impact of mTOR signaling in regulating the morphology and behavior of outer radial glia (oRG) in the developing human cortex.

The reviewers also agree that despite these strengths, there are significant problems that limit the potential impact of the results. In particular, the lack of consideration of other downstream signaling effectors, some that have been previously implicated to contribute to the cell behaviors in the manuscript, and the lack of rigorous statistical methods stand out as the major shortcomings.

Essential revisions:

1) Need for more rigorous statistics, including statistical testing and comparisons as well as presentation of p-values. Specific revisions:

- Several times three or more groups are compared using one-way ANOVA and only one p-value is reported. For normally distributed data one-way ANOVA with multiple comparisons and for not normally distributed data non-parametric test with multiple comparisons should be performed to compare each sample group and show individual p-values. This applies to Figure 1I, Figure 2C, 2D, 2E, Figure 2—figure supplement 2B, Figure 2—figure supplement 2E, Figure 3B, Figure 3—figure supplement 1C, Figure 3—figure supplement 1D, Figure 4A, 4B, Figure 4—figure supplement 1A.

- Figure legends indicate that the mean with standard error of the mean is shown. However, Figure 4B and Figure 3—figure supplement 1D actually show the mean with standard deviation. When reporting individual values, the mean with standard deviation should be given to account for the central tendency and variance of the data. Furthermore, in e.g. Figure 2D or Figure 3—figure supplement 1D and other figures the mean with error bars is not visible as it is covered by data points.

- In Figure 2C the quantification shows mean with standard error of the mean. However, control and rapamycin groups are clearly not normally distributed. Thus, rather the median and quartiles should be used to present the data. Furthermore, one-way ANOVA cannot be applied here as it assumes gaussian distribution of the data. A non-parametric test with multiple comparisons should be used.

- Exact values of N are not given for all figures. This is despite clear statements in the Transparent Reporting file. For example, Figure 4—figure supplement 1A, says "three independent experiments/protein/group", but based on the source file, this figure actually has between 2 and 7 data points.

- While the authors claim organoid experiments where reproduced with 3 independent cell lines there is no data showing this. Individual data points should be shown also in supplementary figures, especially for very low N-numbers and differing N-numbers between samples mean with SEM is misleading.

- The authors state that "slices treated with BDNF/3BDO showed an increase in the number of pS6 expressing cells, indicating activation of mTOR signaling (Figure 2—figure supplement 1)." Figure 2—figure supplement 1A shows no statistical test and based on source data 3BDO shows no significant increase in pS6 cells.

- Individual data points should be shown also in supplementary figures, especially for very low N-numbers and differing N-numbers between samples mean with SEM is misleading.

2) The authors will need to modify some of their conclusions and claims of novelty, taking all the previously published literature into consideration. Specific revisions:

- Modify conclusions: The data in the paper do not rule out other molecular contributors. The authors cannot rule out that the observed rescue in Figure 4 is not also due to activated Rac1 signaling and so the authors should modify their interpretation of this data.

- Modify claims of novelty: The authors demonstrate oRG-specific defects arising from mTOR hyper-activation and inhibition, but both the downstream targets and their effects have been characterized before. In particular, the authors should consider previous studies related to CDC42 (Cappello et al., 2006, Liu et al., 2010 (Rac1)).

3) Developmental timing and developmental disorders. The authors mention that mTOR pathway deregulation is important in several neurodevelopmental disorders. To avoid misinterpretation of the current study and its relevance for disease, acute effects of mTOR pathway deregulation and long-term developmental outcomes should be more clearly distinguished.

[Editors' note: further revisions were suggested prior to acceptance, as described below.]

Thank you for resubmitting your article "mTOR signaling regulates the morphology and migration of outer radial glia in developing human cortex" for consideration by *eLife*. Your revised article has been reviewed by two peer reviewers Anita Bhattacharyya, and the evaluation has been overseen by Marianne Bronner as the Senior Editor.

The reviewers have discussed the reviews with one another and the Reviewing Editor has drafted this decision to help you prepare a revised submission.

For the revision of this manuscript by Andrews et al., authors were asked to modify three aspects of the original results and conclusions: (1) statistical analyses, (2) conclusions and claims of novelty and (3) discussion of developmental disorders. The authors have adequately provided more transparent and rigorous statistical analyses. The Discussion section has been modified to put the results in context of previously published literature and to include more specific consideration of the results in the context of developmental disorders.

However, there are two remaining issues that need to be addressed concerning the presentation of new data on the PSC lines and the way data were changed during the revision so that they no longer support the initial results.

1) The authors state (subsection “Changes in mTOR signaling disrupt the glial scaffold in primary cortical cultures and organoids”) "In the organoid models, there were modest changes to progenitor and neuron proportions in one gain of function condition, with some variability across PSC lines (Figure 2—figure supplement 4;[…])" and the figure supplement is entitled "Effects of mTOR manipulations are generally consistent across PSC lines". Yet, the effects of rapamycin, BDNF and 3BDO treatment differ greatly between the different cell lines, sometimes resulting in opposite effects upon treatments (e.g. Figure 2—figure supplement 3B and Figure 2—figure supplement 4C). Thus, the data show that the effects of mTOR manipulation are not consistent and this should be discussed in the manuscript.

Comparison of treatment effects should be performed within a cell line, ideally presenting the data for control and all treatments next to each other per cell line. Combining the data of all cell lines as done in Figure 2—figure supplement 3B is not informative due to the big differences between cell lines. Rather conditions should be compared within each cell line and subsequently the observed differences should be compared to those in other cell lines. The authors should discuss the observed differences and the limitations arising from these.

2) While the statistics were corrected in the new version of the manuscript, the data used for the figure has also substantially changed. Based on the Source files provided, all the data points above 0.4 were left out and new data were added in the same range as the values that were kept. The same was done for Rapamycin and BDNF, while the 3BDO condition stayed exactly the same. The authors should provide an explanation for why they altered the data in this way. They also should explain how they determined the cellular processes that were measured. This is important as differences between rapamycin, BDNF and 3BDO treatment observed in the previous version are no longer observed in the current version.

---

## [Author Response]

Summary:The reviewers all agree that the manuscript is elegantly performed and provides an important contribution by providing valuable insights into the impact of mTOR signaling in regulating the morphology and behavior of outer radial glia (oRG) in the developing human cortex.The reviewers also agree that despite these strengths, there are significant problems that limit the potential impact of the results. In particular, the lack of consideration of other downstream signaling effectors, some that have been previously implicated to contribute to the cell behaviors in the manuscript, and the lack of rigorous statistical methods stand out as the major shortcomings.Essential revisions:1) Need for more rigorous statistics, including statistical testing and comparisons as well as presentation of p-values. Specific revisions:- Several times three or more groups are compared using one-way ANOVA and only one p-value is reported. For normally distributed data one-way ANOVA with multiple comparisons and for not normally distributed data non-parametric test with multiple comparisons should be performed to compare each sample group and show individual p-values. This applies to Figure 1I, Figure 2C, 2D, 2E, Figure 2—figure supplement 2B, Figure 2—figure supplement 2E, Figure 3B, Figure 3—figure supplement 1C, Figure 3—figure supplement 1D, Figure 4A, 4B, Figure 4—figure supplement 1A.

We thank the reviewers for pointing this out. Normality test (D’Agostino-Pearson) was run for all data sets. If normally distributed, one-way ANOVA with multiple comparisons was performed and all p-values are now reported in figure legends. If not normally distributed, a Kruskall-Wallis Test was performed and p-values reported in figure legends. Comparisons have now been clarified with p-value asterisks (ex. ****p<0.0001) and a line showing comparison between specific treatment group and control.

- Figure legends indicate that the mean with standard error of the mean is shown. However, Figure 4B and Figure 3—figure supplement 1D actually show the mean with standard deviation. When reporting individual values, the mean with standard deviation should be given to account for the central tendency and variance of the data. Furthermore, in e.g. Figure 2D or Figure 3—figure supplement 1D and other figures the mean with error bars is not visible as it is covered by data points.

All graphs with normally distributed data now show the mean with standard deviation. Figures where the error bars were obscured by the data points have now been colored grey, so the mean and standard deviation are visible on top of data points (Figure 2D, Figure 2—figure supplement 3B, Figure 3—figure supplement 1D).

- In Figure 2C the quantification shows mean with standard error of the mean. However, control and rapamycin groups are clearly not normally distributed. Thus, rather the median and quartiles should be used to present the data. Furthermore, one-way ANOVA cannot be applied here as it assumes gaussian distribution of the data. A non-parametric test with multiple comparisons should be used.

The D’Agostino-Pearson normality test was run for all figures, and data in Figure 2C and 2D were found not to be normally distributed. We ran the Kruskall-Wallis non-parametric test to determine significance values. Updated p-values are reported and the median and interquartile range is shown on the graphs.

- Exact values of N are not given for all figures. This is despite clear statements in the Transparent Reporting file. For example, Figure 4—figure supplement 1A, says "three independent experiments/protein/group", but based on the source file, this figure actually has between 2 and 7 data points.

Exact N values are now provided in the figure legends and graphs show individual data points.

- While the authors claim organoid experiments where reproduced with 3 independent cell lines there is no data showing this. Individual data points should be shown also in supplementary figures, especially for very low N-numbers and differing N-numbers between samples mean with SEM is misleading.

The organoid experimental data has now additionally been broken down by stem cell line in Figure 2—figure supplement 4 to demonstrate the reproducibility of observations across PSC lines. Changes to oRG morphology following mTOR manipulations were robust and always reproduced in organoids across iPSC lines. In our cell fate studies, we observed some variability among organoids from different iPSC lines following 3BD0 treatment, in the proportions of Hopx+, Tbr2+ and Ctip2+ cells. However, given the known variability of organoid models (Bhaduri et al., 2020), the modest differences in 3BD0-treated groups and overall consistency in rapamycin and BDNF-treated organoids with our independent observations in primary tissue, we concluded that mTOR manipulations have no significant effect on cell fate. Description of PSC lines used are provided in the Materials and methods section. Individual data points are shown in all supplementary figures and graphs now show standard deviation.

- The authors state that "slices treated with BDNF/3BDO showed an increase in the number of pS6 expressing cells, indicating activation of mTOR signaling (Figure 2—figure supplement 1)." Figure 2—figure supplement 1A shows no statistical test and based on source data 3BDO shows no significant increase in pS6 cells.

We now show statistical significance and report p-values. However, since the number of pS6+ cells in the 3BD0 is not statistically significant, we have modified this claim in the text as follows “slices treated with BDNF/3BDO generally showed increased pS6, indicating activation of mTOR signaling (Figure 2—figure supplement 1).”

- Individual data points should be shown also in supplementary figures, especially for very low N-numbers and differing N-numbers between samples mean with SEM is misleading.

All graphs now show individual data points across figures and figure supplements and show the standard deviation.

2) The authors will need to modify some of their conclusions and claims of novelty, taking all the previously published literature into consideration. Specific revisions:- Modify conclusions: The data in the paper do not rule out other molecular contributors. The authors cannot rule out that the observed rescue in Figure 4 is not also due to activated Rac1 signaling and so the authors should modify their interpretation of this data.- Modify claims of novelty: The authors demonstrate oRG-specific defects arising from mTOR hyper-activation and inhibition, but both the downstream targets and their effects have been characterized before. In particular, the authors should consider previous studies related to CDC42 (Cappello et al., 2006, Liu et al., 2010 (Rac1)).

We have now updated our references and discussion to include previous findings regarding the role of the Rho GTPases on radial glia in the developing mouse cortex. Additionally, we have clarified the possibility that RAC1 may play a role in mediating the changes we observed after mTOR manipulation. However, we observed a more dramatic activation of CDC42 in our Rho GTPase activation assays when compared to RAC1, and the CDC42 response was modulated by rapamycin, while the RAC1 response was not responsive to rapamycin. Therefore, we chose to directly pursue the relationship between mTOR signaling and CDC42. Although we still cannot rule out that RAC1 may play a role in the mTOR response, we want to be conservative in assigning such a role since there is little experimental evidence that it mediates the described phenotype.

This is further described in the Results section as follows: “The Rho GTPases, RHOA, RAC1 and CDC42, are actin regulators and prime candidate molecules to affect radial glial morphology and migration as Rho-ROCK signaling is required for oRG MST divisions (Hansen et al., 2017; Ostrem et al., 2014b) and CDC42 and RAC1 maintain radial glial polarity during mouse cortex development (Yokota et al. 2010; Cappello et al. 2006; Leone et al. 2010)*”* and in the discussion as follows: “Thus, the components of the actomyosin system appear to have complementary roles in regulating different aspects of oRG movement, with Rho-mediated non-muscle myosins regulating MSTs while migration itself is controlled by actin-modulating GTPases, such as CDC42 and RAC1. Effects of CDC42/RAC1 signaling on the morphology, polarity and proliferation of ventricular radial glia (vRG) have been reported in the mouse cortex (Capello et al., 2006; Yokota et al., 2010; Leone et al., 2010), suggesting that the specific activity of mTOR signaling in human oRG cells may control their morphology and migration through conserved molecular pathways.”

3) Developmental timing and developmental disorders. The authors mention that mTOR pathway deregulation is important in several neurodevelopmental disorders. To avoid misinterpretation of the current study and its relevance for disease, acute effects of mTOR pathway deregulation and long-term developmental outcomes should be more clearly distinguished.

We have now expanded our discussion on the implications of our studies and how the described short-term changes to mTOR signaling specifically affect early neural developmental processes. Additionally, we describe the potential long-term impact these changes may have on neuronal structure, maturation, connectivity and ultimately function as follows: “The mTOR signaling pathway is dysregulated in many neurodevelopmental disorders including autism and Fragile X Syndrome, as well as in cortical malformations such as Focal Cortical Dysplasia and Tuberous Sclerosis. These complex neurodevelopmental disorders arise as a result of accumulated errors throughout development affecting both proliferative cells as well as the migration, morphology, connectivity and plasticity of neurons and glia (Subramanian et al., 2020). Discrete short-term and long-term changes to mTOR signaling can therefore create dynamic structural and functional changes in the brain, complicating our understanding of these diseases. A thorough characterization of these different effects of mTOR dysregulation in human cortical development is therefore crucial to the creation of better disease models and therapies for multiple neurodevelopmental disorders.”

[Editors' note: further revisions were suggested prior to acceptance, as described below.]

For the revision of this manuscript by Andrews et al., authors were asked to modify three aspects of the original results and conclusions: (1) statistical analyses, (2) conclusions and claims of novelty and (3) discussion of developmental disorders. The authors have adequately provided more transparent and rigorous statistical analyses. The Discussion section has been modified to put the results in context of previously published literature and to include more specific consideration of the results in the context of developmental disorders.However, there are two remaining issues that need to be addressed concerning the presentation of new data on the PSC lines and the way data were changed during the revision so that they no longer support the initial results.1) The authors state (subsection “Changes in mTOR signaling disrupt the glial scaffold in primary cortical cultures and organoids”) "In the organoid models, there were modest changes to progenitor and neuron proportions in one gain of function condition, with some variability across PSC lines (Figure 2—figure supplement 4;[…])" and the figure supplement is entitled "Effects of mTOR manipulations are generally consistent across PSC lines". Yet, the effects of rapamycin, BDNF and 3BDO treatment differ greatly between the different cell lines, sometimes resulting in opposite effects upon treatments (e.g. Figure 2—figure supplement 3B and Figure 2—figure supplement 4C). Thus, the data show that the effects of mTOR manipulation are not consistent and this should be discussed in the manuscript.Comparison of treatment effects should be performed within a cell line, ideally presenting the data for control and all treatments next to each other per cell line. Combining the data of all cell lines as done in Figure 2—figure supplement 3B is not informative due to the big differences between cell lines. Rather conditions should be compared within each cell line and subsequently the observed differences should be compared to those in other cell lines. The authors should discuss the observed differences and the limitations arising from these.

We appreciate the reviewer’s suggestions and have now clarified changes observed within and across individual stem cell lines. As noted by reviewers, there is variability in proportions of cell types across different stem cell lines. However, the results indicate minimal changes to cell fate overall. We have now represented cell fate organoid data in Figure 2—figure supplement 4 only and show the findings by cell line in detail. Figure 2—figure supplement 3 is now solely focused on the cell fate results from the primary tissue studies. Graphs in Figure 2—figure supplement 4 now represent each cell line with the control and treatment conditions together and statistics run within each group. Changes to particular markers in a given cell line and condition are detailed in the figure legend. The text in the manuscript now reads “In the organoid models, there were modest changes to progenitor and neuron proportions in some mTOR manipulation conditions. However, these results were inconsistent across PSC lines and conditions. Therefore, the organoid model may not be a reliable indicator of mTOR-induced changes in cell number (Figure 2—figure supplement 4; n>34sections/group, six organoids/group from three experiments). Moreover, the results did not reflect the observations from slice culture experiments, a more cytoarchitecturally accurate model of human cortex development”.

2) While the statistics were corrected in the new version of the manuscript, the data used for the figure has also substantially changed. Based on the Source files provided, all the data points above 0.4 were left out and new data were added in the same range as the values that were kept. The same was done for Rapamycin and BDNF, while the 3BDO condition stayed exactly the same. The authors should provide an explanation for why they altered the data in this way. They also should explain how they determined the cellular processes that were measured. This is important as differences between rapamycin, BDNF and 3BDO treatment observed in the previous version are no longer observed in the current version.

We appreciate the opportunity to address this difference in our re-submitted manuscript primary slice culture data for Figure 2C. In our initial submission, we had included all slice culture data that had been collected and analyzed over many experiments that spanned several years, due to limited access of primary human tissue. However, upon re-analysis of our data, per reviewers’ comments, we realized that some of the initial studies did not include all mTOR conditions in every experiment. Although the trend for the early dataset was consistent, with a clear decrease in fiber length after mTOR manipulation, we decided it would be best to only include full experiments that had all mTOR conditions along with matched, paired controls and had been quantified using consistent, well-defined criteria. We apologise for not clarifying this change in our first revision. For full transparency, these initial datasets are now represented in Figure 2—figure supplement 1D and exclusion criteria is now detailed in the transparent reporting form.

Measurement of cellular processes: For all conditions, we identified the GFP+ cell in the oSVZ with the longest basal process, along with two additional representative cells for a total of three GFP+ processes per slice culture. The length of the basal process in these cells was measured between its point of origin on the basal side of the cell and its end using Imaris software as described in the subsection “Primary Basal Fiber Length Quantification”. The height of the slice is measured from the edge of the ventricular zone to the edge of the cortical plate. The GFP fiber length is then divided by the total slice height to determine the proportional fiber length. The quantification methods are additionally detailed in the Materials and methods section.